# Damped Cantilever Microprobes for High-Speed Contact Metrology with 3D Surface Topography

**DOI:** 10.3390/s23042003

**Published:** 2023-02-10

**Authors:** Michael Fahrbach, Min Xu, Wilson Ombati Nyang’au, Oleg Domanov, Christian H. Schwalb, Zhi Li, Christian Kuhlmann, Uwe Brand, Erwin Peiner

**Affiliations:** 1Institute of Semiconductor Technology (IHT), Technische Universität Braunschweig, Hans-Sommer-Straße 66, 38106 Braunschweig, Germany; 2Laboratory for Emerging Nanometrology (LENA), Langer Kamp 6a/b, 38106 Braunschweig, Germany; 3Physikalisch-Technische Bundesanstalt (PTB), Bundesallee 100, 38116 Braunschweig, Germany; 4Department of Metrology, Kenya Bureau of Standards (KEBS), Popo Rd, Nairobi 00200, Kenya; 5GETec Microscopy GmbH, Am Heumarkt 13, 1030 Vienna, Austria; 6Quantum Design Microscopy GmbH, Im Tiefen See 60a, 64293 Darmstadt, Germany

**Keywords:** piezoresistive cantilever, tactile surface scanning, high-throughput metrology, contact mode, resonance frequency, quality factor

## Abstract

We addressed the coating 5 mm-long cantilever microprobes with a viscoelastic material, which was intended to considerably extend the range of the traverse speed during the measurements of the 3D surface topography by damping contact-induced oscillations. The damping material was composed of epoxy glue, isopropyl alcohol, and glycerol, and its deposition onto the cantilever is described, as well as the tests of the completed cantilevers under free-oscillating conditions and in contact during scanning on a rough surface. The amplitude and phase of the cantilever’s fundamental out-of-plane oscillation mode was investigated vs. the damping layer thickness, which was set via repeated coating steps. The resonance frequency and quality factor decreased with the increasing thickness of the damping layer for both the free-oscillating and in-contact scanning operation mode, as expected from viscoelastic theory. A very low storage modulus of $E^{\prime }\approx 100\;\mathrm{kPa}$, a loss modulus of $E^{\prime \prime }\approx 434\;\mathrm{kPa}$, and a density of $\rho \approx 1.2\;g\;{\mathrm{cm}}^{-3}$ were yielded for the damping composite. Almost critical damping was observed with an approximately 130 µm-thick damping layer in the free-oscillating case, which was effective at suppressing the ringing behavior during the high-speed in-contact probing of the rough surface topography.

## 1. Introduction

In recent years, the role of manufacturing metrology has significantly changed towards in-process monitoring where all relevant knowledge regarding the quality of a workpiece is obtained while it is being manufactured [1]. Such a closed-loop manufacturing process can increase production performance, reduce costs, and improve product quality through the combination of digital technologies, manufacturing, and measuring operations based on inspection and consumer feedback. To measure the geometrical and dimensional features, non-contact optical metrology solutions are commercially available and are advantageously utilized due to their inherent speed in high-throughput measurements. However, reflective, transparent, or dark objects are not suitable for optical measurements. Furthermore, modern workpieces, e.g., produced by additive manufacturing, commonly exhibit complex free-form geometries and hollow shapes (e.g., inaccessible narrow holes), as well as a mix of random and deterministic surface features. In metals, surface features such as a large range of scales of interest, step-like transitions, overhangs, and highly reflective and opaque surface regions can cause significant difficulties for measurement instruments, while material translucency is an additional difficulty for polymer surfaces.

The surface topography denotes the deviation of a real surface from an ideal flat plane, due to flaws, form error, waviness, and roughness [2]. The evaluation of these surfaces has been changing from a 2D profile to 3D area characterization, i.e., the determination of the surface roughness and surface waviness has been extended to the surface texture and surface form evaluation. The surface texture can be predicted optically from captured surface images. However, the image quality for machine vision using natural light sources is influenced by the non-uniform illumination, low depth of focus, and noise sources (e.g., dirt, cut chips, etc.) in industrial environments. Other optical techniques such as phase-shifting interferometry or coherence scanning interferometry usually have only a small field of view, i.e., subtle textural changes cannot be revealed across the entire surface of a workpiece. High-definition metrology based on laser triangulation can measure the three-dimensional (3D) surface topography of an engine block with a 0.15 mm resolution and 1 µm accuracy in the lateral and depth directions, respectively [2]. To achieve a much better lateral and vertical resolution close to 0.1 µm and 0.05 nm, respectively, required for tool condition monitoring, it is necessary to use stylus techniques, which measure the surface texture, such as roughness and waviness in contact. For this task, conical diamond microprobes of a radius between 0.1 µm and 10 µm and a cone angle between 60° and 90° are commonly employed. Unfortunately, so far, contacting stylus techniques generally only deliver localized information under offline conditions.

The most-important drawback of contact measurement methods is, thus, the extended time that is necessary for the 3D measurements of a surface’s topography. Here, piezoresistive silicon cantilever microprobes offer the potential for much faster surface scanning at traverse speeds up to $15\;\mathrm{mm}\;s^{-1}$ while providing a high vertical and lateral resolution, better than 10 nm and better than 1 µm, respectively, maintained within a vertical depth range greater than 0.1 mm [3]. These microprobes integrate a silicon probing tip at the free end of the cantilever and a piezoresistive Wheatstone bridge as a strain gauge at its clamped end. They can measure form and roughness inside of narrow holes (for example, the spray holes of fuel injection nozzles of 100 µm in diameter and 1 mm in depth [4,5]), as well as, for example, on paper-making machine rolls of a diameter of 1 m and a length of 8 m over a scan length of 100 mm in industrial settings [6]. Operated with an external piezoactuator at the cantilever base, they can also image mechanical surface properties using Dual AC Resonance Tracking (DART) in a commercial Asylum Cypher AFM [7]. Furthermore, a cantilever-based measurement unit has been reported, which has a volume of only a few ${\mathrm{mm}}^{3}$. This unit can be mounted in a skid body, forming a miniature, battery-operated roughness tester with Bluetooth data transmission to a computer for autarkic measuring on the workpiece, e.g., in bores of a length of up to 12.5 mm, at a speed up to $5\;\mathrm{mm}\;s^{-1}$, and in a vertical range of 0.1 mm. The cantilever has a full diamond tip with a 2 µm radius and a cone angle of 90° [3].

The scanning speed of a contact profilometer is limited by the trackability of the surface topography [8]. Owing to their high dynamics (determined from a fundamental flexural resonance frequency of $f_{0}\approx 3\;\mathrm{kHz}$ at a quality factor of Q≈600, cf. Table 1), cantilever microprobes maintain the trackability of step-like features (with inclination variations of up to 70°) during high-speed contact probing of at least $10\;\mathrm{mm}\;s^{-1}$ [9].

Under such high-speed conditions, however, the integrated silicon tip is affected by substantial wear [11] and was, therefore, replaced by a full monocrystalline diamond probing tip glued to the bottom side of the cantilever [9]. While the dynamics of this probe remain high enough to ensure trackability, it shows oscillating behavior during contact profilometry at a probing force of F≈100μN and a speed of $5\;\mathrm{mm}\;s^{-1}$ [9]. The larger contact stiffness due to the larger tip size (height of 210 µm and diameter of 2 µm) and mass of the diamond tip (12 µg) vs. the silicon tip (height of 100 µm, diameter of 0.1 µm, and mass of 0.15 µg) may induce stick–slip effects during scanning.

Furthermore, microprobes with diamond tips show oscillations of the cantilever, which disturb the 3D topography imaging performed using line-by-line scanning. Here, it can take a tens of micrometers scan path until the force–control loop of the setup can maintain the tip in contact with the sample surface. Figure 1 shows such disturbances in the surface topography image (300μm×300μm) of a silicon micropillar array (4 µm in height, 1.2 µm in diameter, 4 µm in pitch) measured by a microprobe with a diamond tip.

A higher tracking bandwidth will be necessary to avoid this effect, which can be achieved by increasing the resonance frequency or by reducing its Q factor. The microprobe geometry would have to be changed (i.e., reduction of length and/or increase of thickness), which is not desirable to access high-aspect-ratio topography. Here, we therefore concentrated on increasing the damping. This will also be beneficial to avoid the coupling of ground vibrations or acoustic noise in the microcantilever generated by machines in an industrial environment.

In this work, commercial cantilever sensors (CAN50-2-5, CiS Forschungsinstitut für Mikrosensorik GmbH, Erfurt, Germany) were used, whose dimensions and layout are shown in Table 2 and Figure 2a, respectively.

A comprehensive overview of the methods to tune the Q factor of MEM/NEM resonators was given in [12], and the selection of the approaches for damping cantilever sensors is listed in Table 3.

The best results were achieved using closed-loop *Q* control. Here, a force proportional to the oscillation velocity of the beam was applied to the sensor. Although measuring velocity [27,38] is preferable, methods based on measuring deflection, such as differentiation [13], phase-shifting [20], time delay [16], resonant control [22], observer-based control [39], and positive position feedback [24,26], are also viable. In general, the effectiveness of closed-loop *Q* control decreases with the increasing integration level of the sensor. Nevertheless, when used with a self-sensing and self-actuating design, it improves the response time significantly, which enables accurately tracking the topography of step-like features on a calibration grating at higher scan rates [26,40].

If no sufficient feedback or controller is available, open-loop control can also be used. For tapping-mode measurements, the damping can be increased by applying a frequency-dependent phase shift to the actuation signal of the tapping-piezo and supplying it to a second piezoactuator [28]. Alternatively, optical pumping [29,30,31] and mechanical pumping [32,33] enable dampening the resonant oscillations by coupling two vibration modes. However, as these methods are tuned to the resonant frequencies of the beam, they are best suited for measurements during which these frequencies remain constant (which is not the case for contact mode measurements).

Finally, passive damping can be employed to decrease the Q factor. If the sensor has a piezoelectric actuator integrated with it, this can be shunted with an electrical impedance to decrease the Q factor of the sensor by creating a damped resonant circuit [34]. However, this method has the same drawbacks as open-loop control. Alternatively, high-loss materials can be used to increase the mechanical losses [35,36,37].

## 2. Materials and Methods

Unfortunately, the CAN50-2-5 microprobe does not contain an integrated actuator that could be used for *Q* control. An external actuator could be used instead; however, this increases the amount of instrumentation required to operate the device. As an alternative, we therefore considered polymer coatings, which are deposited on cantilever-based chemical sensors for trapping specific analytes. They show a damping effect on the Q factor of the cantilever due to their viscoelastic properties [41]. Dip-coating of a silicon cantilever with dimensions similar to the CAN50-2-5 (length: 5 mm, width: 200 µm, thickness: 40 µm) with photoresist resulted in a moderate reduction of the Q factor in air from 502 to 398; however, the probing tip was coated, as well [36]. Furthermore, the stability of the standard photoresist is not sufficient for a device that is operated in an industrial environment, where it may be exposed over long periods of time to lubricants, solvents, heat, and radiation. These issues were overcome by using epoxy-based high-loss materials, which have already been employed to develop high-speed sensors [37]. In this work, we, therefore, employed a mixture of epoxy adhesive (UHU® quickset, UHU GmbH & Co. KG, Bühl, Germany), isopropyl alcohol (IPA), and glycerol (EIG). This glue is resistant to the relevant chemicals and moderate exposure to UV radiation. Additionally, it is stable at temperatures in the range of −40 °C to 100 °C. When mixed with glycerol, the epoxy remains viscous after curing, probably due to the plasticizing or emulsifying properties of glycerol. To dilute the mixture for tip dispensing and to enable a smooth coating layer, IPA was added. It eventually evaporates within a few days after deposition.

Initial tests of the EIG coating were performed using a piezoresistive AFM cantilever (AMGT, Botevgrad, Bulgaria), showing a drastic decrease of the Q factor from 451 to 53 in air (Figure 3). With this cantilever operated in an AFSEM^®^ (GETec, Vienna, Austria), the step-like features of the calibration grating were imaged at $33\;\mathrm{lines}\;s^{-1}$ corresponding to a scanning velocity of $1\;\mathrm{mm}\;s^{-1}$. The EIG coating was effective at damping the resonant oscillation of the AFM cantilever and was thus selected in this study for the application with the CAN50-2-5 microprobes, as well.

### 2.1. Coating Preparation

Both components of the UHU® quickset epoxy glue were mixed according to the datasheet instructions. Afterwards, glycerol and IPA were mixed sequentially. This EIG mixture was then decanted into a 3 mL syringe cartridge and sonicated for five minutes. Finally, the sediments were removed, and a dispensing tip with an inner diameter of 0.1 mm was screwed into the cartridge of the syringe.

To achieve the optimal damping characteristics, we maximized the content of the glycerol. This was realized at a volume ratio of epoxy glue to IPA to glycerol of 4:20:2. Higher contents of glycerol cannot be stably emulsified with this method in an EIG mixture. The EIG mixture of 4:20:1 was used (on one cantilever with $t_{2}=69\;\mu m$) to evaluate the effect of a deviation from the maximum glycerol content on the damping.

The epoxy:IPA:glycerol mixture (EIG) concentrations after deposition on the cantilever can differ from the nominal values, as the components of higher density may be removed as sediments or after decanting. Additionally, after loading into the syringe, the component with the lowest density, i.e., IPA, may float to the top of the mixture and would, thus, not be deposited on the cantilever. Therefore, we expected that the compositions of the EIG coatings with different glycerol contents (ratios of 4:20:2 and 4:20:1 of epoxy:IPA:glycerol) would differ less than the numerical values may suggest.

### 2.2. Coating Deposition

To coat the CAN50-2-5 microprobes with the polymer emulsion, a computer-controlled dispenser setup was employed [42]. We used an extrusion-based method developed for 3D printing of bioinspired materials known as pressure-assisted micro syringe (PAM) printing [43]. A syringe was connected to the dispenser and mounted on an xyz-positioning stage with a resolution of 10 µm in all axes. As shown in Figure 1, the sensor was placed beneath the syringe, which can be pressurized controllably in the range of 0 bar to 1 bar for durations of 1 ms to 99.9 s. By simultaneously operating the dispenser and the stages, volumes down to 50 pL can be deposited at the selected position on the cantilever.

For every coating step, the syringe was initially moved to a position above the deflection-sensing strain gauge (Wheatstone bridge) of the sensor. Starting from here, the syringe was pressurized to 0.15 bar for 10 ms at ten equidistant points along the cantilever axis. This was performed twice to improve the homogeneity of the coating layer. A video of a single step is included in the Appendix A. Nevertheless, a significant amount of liquid did not stick to the cantilever, but was pulled upwards along the dispenser needle. This was especially noticeable during the first deposition. Additionally, the creep of the liquid towards the cantilever base was observed, which covered the contact pads of the Wheatstone bridge, as shown in Figure 4.

This excess material was removed after the final coating/curing step. Between steps, the thickness of the coating was determined using an optical microscope. Finally, the coating was cured for approximately 24 h.

The average thickness is related to the number of deposition steps and the volume ratio of the components of the coating. It increased by approximately 45% per step. The evolution of the coating of a single sensor over ten deposition steps is shown in the Appendix A. Its (absolute) standard deviation, which was used as a measure of the uniformity of deposition, increased with every deposition step. However, its relative value remained between 40% and 50% of the average thickness at coating thicknesses of 40 µm and above. A much better controllability of the deposition process can be concluded from the resonance frequencies measured with four different cantilevers of the same number of coatings, i.e., a thickness of 12 µm ± 12 µm (cf. Table 1 and Figure 5). Here, we found a resonance frequency averaged over the four devices of 3061 Hz ± 33 Hz, whose standard deviation indicated a relative error of only ±1.1%. This was in the range of the expected deviations between different uncoated sensors of a cantilever-fabrication batch.

### 2.3. Measurement and Analysis Procedures

After coating, the cantilever microprobes were wire-bonded to different printed circuit boards (PCBs) developed by TU-BS and PTB. The TU-BS PCB was designed for contact resonance measurements and is equipped with a preamplifier (THS4131 and OPA1612, Texas Instruments Incorporated, Dallas, TX, USA) and a chip piezoactuator (5mm×5mm×2mm, PL 055.30 PICMA® Chip Actuator, PI Ceramic, Lederhose, Germany), which was placed between the base of the microprobe and the PCB to excite the cantilever in its fundamental out-of-plane resonance mode. The output ports of the preamplifiers were connected to an external data acquisition system. The PTB PCB was designed for high-speed form and roughness measurements and does not contain any components in addition to the sensor. An external actuator was placed under the holder of the microprobe for resonance actuating. The piezo-resistive strain gauge was connected to a strain gauge amplifier (ML10B and MGCplus, Hottinger, Brüel & Kjær, Darmstadt, Germany). Long-term measurements with cantilevers having coatings in the range of 10 µm to 130 µm thick were performed after aging up to almost one and a half years (cf. Table 1) to validate the viscoelastic stability of the coating. The measured frequency response of a selection of sensors, respectively, is shown in Figure 5.

Scanning measurements in contact on sandpaper (with a grain size of approximately 0.3 µm) were carried out using the EIG-coated cantilevers (and an uncoated cantilever as a reference) at a constant probing force of F≈50μN. For this, a self-developed surface topography profiler (Profilscanner, PTB, Braunschweig, Germany) was employed [3,9], which consisted of a xyz piezo stage (P-628.2CD for the xy axes and P-622.ZCD for the *z* axis, Physik Instrumente (PI) GmbH & Co.KG, Karlsruhe, Germany) with a motion range of 800μm×800μm×250μm (x×y×z).

The damping effect of the coatings was evaluated based on the measured frequency dependences of the amplitude and phase by fitting the output voltage *U* in the complex plane using a Fano line shape function (FLSF):(1)$$U=\frac{\frac{U_{0}}{Q}e^{i\varphi_{\mathrm{del}}}}{{\left(i\frac{\omega }{\omega_{0}}\right)}^{2}+i\frac{\omega }{Q\omega_{0}}+1}+U_{\mathrm{cou}}e^{i\varphi_{\mathrm{cou}}}.$$

Here, $U_{0}$ is the peak amplitude in resonance, ω=2πf is the angular frequency, $\omega_{0}=2\pi f_{0}$ is the angular resonance frequency, and *Q* is the quality factor of the resonant oscillation. The coefficient $\varphi_{\mathrm{del}}$ describes a phase delay in the measurement, and $U_{\mathrm{cou}}$ and $\varphi_{\mathrm{cou}}$, respectively, denote the magnitude and phase delay of coupled signals induced by parasitic crosstalk between the excitation and detection elements on the cantilever sensor, as well as between the respective transmission lines [44]. $U_{0}$, *Q*, $\varphi_{\mathrm{del}}$, $U_{\mathrm{cou}}$, and $\varphi_{\mathrm{cou}}$ are the fitting parameters.

## 3. Results

The effect of EIG deposition on the dynamical behavior of CAN50-2-5 microprobes was investigated under free-oscillation conditions and by scanning on sandpaper with a grain size of approximately 0.3 µm.

### 3.1. Q Factor of Free-Oscillation

The resonance frequencies and Q factors of the fundamental out-of-plane mode measured with uncoated and EIG-coated cantilevers are given in Table 1. Measurements with control samples of one coating showed long-term stable viscoelasticity, as indicated by the negligible change in Q factor over a period of 15 months. The influence of the weight and viscoelasticity of the polymer mixture on the dynamic behavior is shown in Figure 6, where the resonance frequency and quality factor decreased with increasing coating layer thickness $t_{2}$.

The error bars revealed the large inhomogeneity of the layer thickness, which drastically increased with the number of coating steps. Furthermore, the coating should not be thicker than necessary for damping since this will not yield better measurement results. On the contrary, thicker coatings decrease the resonance frequency of the sensor and, thereby, its dynamic range, i.e., maintaining of contact during 3D topography scanning at a high traverse speed.

### 3.2. Scanning on Sand Paper

Scanning experiments on sandpaper were performed with cantilevers coated with EIG of different coating thicknesses at a probing force of F≈50μN and traverse speeds of $20\;\mu m\;s^{-1}$, $1\;\mathrm{mm}\;s^{-1}$, $5\;\mathrm{mm}\;s^{-1}$, and $10\;\mathrm{mm}\;s^{-1}$. Figure 7a shows a photograph of the used roughness artifact (sandpaper of approximately a 0.3 µm grain size), and Figure 7b shows the detail of a surface scan with the CAN50-2-5 microprobe at a probing force of F≈50μN and a traverse speed of $10\;\mathrm{mm}\;s^{-1}$.

Superimposed on the surface topography, a periodic oscillation is visible at 13.62 kHz as determined by fitting Equation (Equation 1) to the corresponding amplitude–frequency characteristics in Figure 7c which was obtained by Fourier transformation. This frequency corresponds to 4.3-times the free-oscillation resonance frequency of the coated cantilever, which was expected for the fundamental flexural contact resonance frequency [45,46,47]. In our case, the contact resonance frequency $f_{0},c$ was actuated by the dynamic forces on the tip, which were induced during its scanning on the sandpaper, which can be described as a white noise source. Correspondingly, the oscillation amplitude was observed to increase proportionally to the scanning speed.

The elastic interaction between the probing tip and sample was modeled with the contact stiffness $k_{c}$, which, according to the Hertzian contact theory, is given by [45,46,47]
(2)$$k_{c}=\sqrt[{3}]{6RFE_{r}^{2}},$$
with the tip radius *R*, the probing force *F*, and the reduced elastic modulus $E_{r}$. The larger value of Young’s modulus and the twenty-times larger tip radius of the diamond tip vs. the silicon tip (Table 1) led to a much higher contact stiffness. Thus, this increased the susceptibility with respect to the oscillating behavior of the cantilever. Similarly, the higher bending moment and shear force on the cantilever owing to the larger height and weight of the diamond tip may lead to oscillations under free-flight conditions, e.g., due to the acceleration force on the tip in the course of line-by-line scanning (Figure 1). Furthermore, the contact resonance peak in Figure 7 is broader than the free-oscillating resonance peak in Figure 1. The contact quality factor of $Q_{c}\approx 30\pm 17$ (which was determined from measurements at traverse speeds of $1\;\mathrm{mm}\;s^{-1}$, $5\;\mathrm{mm}\;s^{-1}$ and $10\;\mathrm{mm}\;s^{-1}$) indicates higher damping with respect to Q=81±8.

All scanning experiments on the sandpaper, which were performed with cantilevers coated with EIG of different coating thicknesses, were evaluated as described above. The values of the contact resonance frequency, contact quality factor, and contact deflection amplitude per scanning speed decreased with the coating layer thickness, as shown in Figure 8.

We found a total reduction of the resonance frequency by 50% and 25% at free-oscillation and in contact, respectively, and the quality factor decreased by 99.8% and 75% at free-oscillation and in contact, respectively.

## 4. Discussion

The mass increase and viscoelastic losses of the cantilever by the polymeric coating were expected to be the primary factors responsible for the decrease in the resonance frequency and quality factor. These reductions increased as the coating thickness increased. In a cantilever composed of two layers (indicated in the following by the indices 1 and 2), the resonance frequency and quality factor are given by [41]
(3)$$f_{0}=\frac{1.875^{2}}{2\pi L^{2}}\sqrt{\frac{E_{1}I_{1}+E_{2}^{\prime }I_{2}}{m_{L}}}$$
(4)$$Q\approx \frac{1}{2\left(1-\sqrt{1-\frac{E_{2}^{\prime \prime }I_{2}}{E_{1}I_{1}+E_{2}^{\prime }I_{2}}}\right)}$$
with the area moments of inertia:(5)$$I_{1}=\frac{bt_{1}^{3}}{12}+bt_{1}{\left(t_{n}-t_{2}-\frac{t_{1}}{2}\right)}^{2}\;\mathrm{and}\;I_{2}=\frac{bt_{2}^{3}}{12}+bt_{2}{\left(t_{n}-\frac{t_{2}}{2}\right)}^{2}$$
the position of the neutral axis:(6)$$t_{n}=\frac{t_{2}}{2}+\frac{t_{1}E_{1}}{2}\frac{\left(t_{1}+t_{2}\right)\left(t_{1}E_{1}+t_{2}E_{2}^{\prime }\right)}{{\left(t_{1}E_{1}+t_{2}E_{2}^{\prime }\right)}^{2}+{\left(t_{2}E_{2}^{\prime \prime }\right)}^{2}}$$
and the composite cantilever mass per unit length:(7)$$m_{L}=\rho_{1}t_{1}+\rho_{2}t_{2}.$$

Here, $E_{1}=170\;\mathrm{GPa}$ is Young’s modulus of silicon, $E_{2}^{\prime }+iE_{2}^{\prime \prime }$ is the complex Young’s modulus of the viscoelastic coating layer, and $\rho_{1}=2.33\;g\;{\mathrm{cm}}^{-3}$ and $\rho_{2}$ are the densities of silicon and the viscoelastic coating layer, respectively. The storage modulus, loss modulus, and density of the coating layer were determined by fitting Equations (Equation 3) and Equation 4 to the measured dependences of the resonance frequency and Q factor in Figure 6. This yielded $E_{2}^{\prime }\approx 100\;\mathrm{kPa}$, $E_{2}^{\prime \prime }\approx 434\;\mathrm{kPa}$ (i.e., loss tangent $E_{2}^{\prime \prime }/E_{2}^{\prime }\approx 4.3$) and $\rho_{2}\approx 1.2\;g\;{\mathrm{cm}}^{-3}$. The density corresponded well to the expectation for polymers and elastomers, while the storage modulus and loss tangent were considerably beyond the expected range of collated elastomer data ($E_{2}^{\prime }>1\;\mathrm{MPa}$, $E_{2}^{\prime \prime }/E_{2}^{\prime }<3$) [48].

The property of EIG of a storage modulus smaller than the loss modulus was not beneficial here, as it provided a damping layer that did not hold its shape during extrusion through the dispenser tip (Figure 4) [43]. Rather, the liquid-like EIG suspension tended to be pulled back along the outer surface of the tip, especially during the first deposition, as well as to creep towards the cantilever base, thereby covering the contact pads. The composition of the EIG will have to be optimized, e.g., by adjusting the content of glycerol. Simultaneously, however, the high mechanical loss properties of the EIG layer for efficient damping of a microcantilever probe shall not be affected.

## 5. Conclusions

The coating of a cantilever microprobe using an emulsion of epoxy glue, isopropyl alcohol, and glycerol was effective at damping its fundamental resonance mode during tactile 3D surface topography measurements. We found a decreasing resonance frequency and quality factor of the free-oscillating cantilever with increasing damping layer thickness. The final coating layer thickness of $t_{2}\approx 130\;\mu m$ enabled us to decrease the Q factor to 1.5, which resulted in a Q factor reduction ratio of $25\times 10^{-4}$. Comparable values have previously only been achieved using active closed-loop *Q* control with external sensor feedback.

According to the theoretical prediction for viscoelastic damping, we calculated values of $E_{2}^{\prime }\approx 100\;\mathrm{kPa}$, $E_{2}^{\prime \prime }\approx 434\;\mathrm{kPa}$, and $\rho_{2}\approx 1.2\;g\;{\mathrm{cm}}^{-3}$ for the storage modulus, loss modulus, and density of the coating layer material. The corresponding curves were found for the contact resonance (CR) frequency and quality factor during surface topography measurement. This CR signal, which adversely affected the in-contact roughness profile measurement of the sandpaper, was nearly entirely suppressed at the final coating layer thickness. In future work, the EIG composition will be optimized towards a larger storage modulus for better shape holding of the layer after dispensing on the cantilever. Furthermore, cantilevers with integrated actuators (either electrothermal or piezoelectric) will be investigated to evaluate active *Q* control with respect to the damping layer concept of the present study.

## Figures and Tables

**Figure 1 sensors-23-02003-f001:**
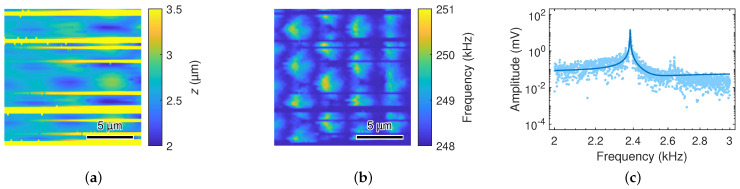
(**a**) Surface topography and (**b**) contact resonance image, respectively, of a silicon micropillar array (pillar height: 4 µm, pillar diameter: 1.2 µm, pillar pitch: 4 µm) acquired using line-by-line scanning of an area of 300μm×300μm (lateral resolution: 0.25 µm, contact force: 6 µN, traverse speed: $0.2\;\mathrm{mm}\;s^{-1}$). (**c**) Oscillations at 2.38 kHz are observed in the Fourier-transformed output signal (points by measurement, line by fitting) during the free-flight positioning of the tip from the end of a scan to the next row, leading to irregular disturbances in the surface topography at the beginning of individual lines.

**Figure 2 sensors-23-02003-f002:**
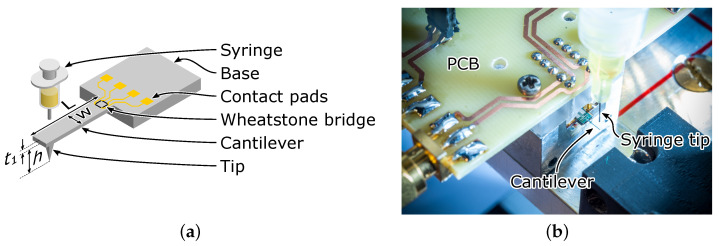
(**a**) Schematic of the silicon cantilever microprobe. (**b**) A computer-controlled tip-dispensing setup is used to deposit droplets of the damping solution onto the cantilever. Photo commissioned by TU Braunschweig and taken by Jan Hosan.

**Figure 3 sensors-23-02003-f003:**
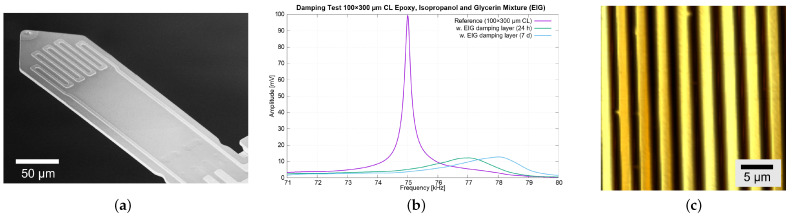
(**a**) Scanning electron microscopy (SEM) photograph of a piezoresistive AFM cantilever (100μm×300μm CL), (**b**) amplitude spectra around its free fundamental resonance frequency before and after the deposition of EIG (Q=451 and 53, respectively), and (**c**) 34μm×34μm image of the line pattern taken with the cantilever in an AFSEM^®^ (GETec, Vienna, Austria) at $33\;\mathrm{lines}\;s^{-1}$, corresponding to a traverse speed of $1\;\mathrm{mm}\;s^{-1}$.

**Figure 4 sensors-23-02003-f004:**
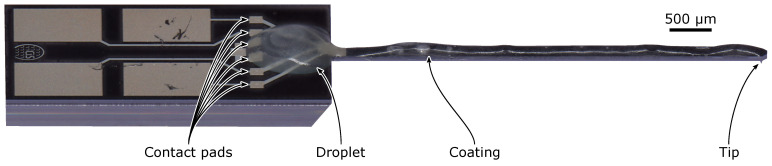
Photo of a microprobe with a coating of 40 µm ± 17 µm in thickness. The photo was taken at an angle of 60°.

**Figure 5 sensors-23-02003-f005:**
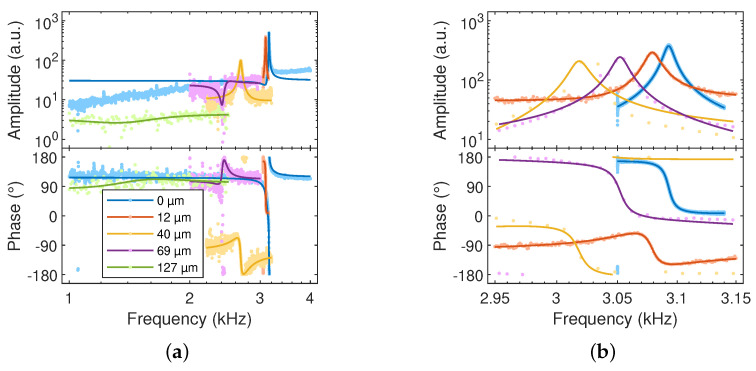
(**a**) Normalized frequency response of cantilevers with different coating thicknesses. (**b**) Four different sensors with the same number of depositions corresponding to a coating thickness of 12 µm ± 12 µm were individually analyzed to evaluate the reproducibility of the coating process. The determined values of the resonance frequency and quality factor are given in Table 1.

**Figure 6 sensors-23-02003-f006:**
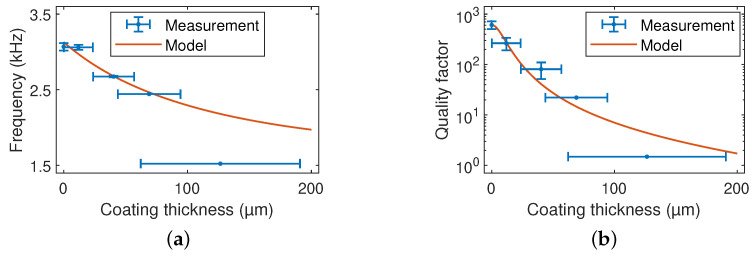
(**a**) Resonance frequency and (**b**) quality factor with dependence on the coating layer thickness. The superimposed lines were calculated using an analytical model described in Section 4.

**Figure 7 sensors-23-02003-f007:**
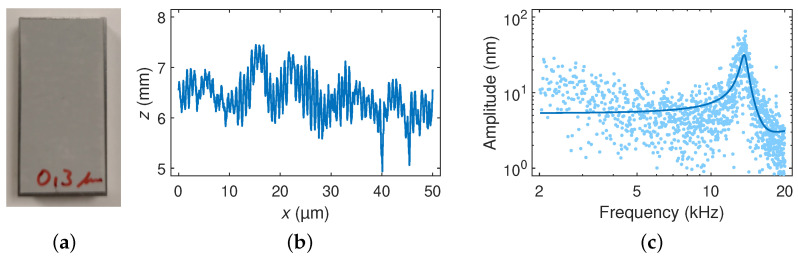
(**a**) Sandpaper of a 0.3 µm grain size and (**b**) surface profile, as well as (**c**) amplitude spectrum taken with the CAN50-2-5 microprobe at $10\;\mathrm{mm}\;s^{-1}$. The light blue dots show the Fourier transform of the time-dependent output signal of the sensor. The dark blue line shows a resonance fit. The amplitude spectrum shows the surface topography of the sandpaper modulated by a contact resonance frequency of $f_{0},c=13.62\;\mathrm{kHz}$ at a quality factor of $Q_{c}\approx 16$ of the cantilever caused by the moving tip in contact with the sample (probing force F≈50μN). In these measurements, a Bessel low-pass filter with a cutoff frequency of 100 kHz was employed, i.e., amplitude suppression was not expected.

**Figure 8 sensors-23-02003-f008:**
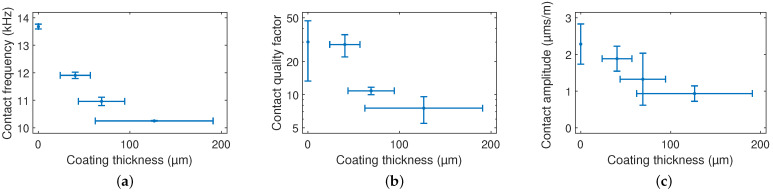
(**a**) Contact resonance frequency $f_{0},c$, (**b**) contact quality factor $Q_{c}$, and (**c**) contact deflection amplitude per scanning speed with dependence on the coating layer thickness $t_{2}$, respectively.

**Table 1 sensors-23-02003-t001:** Free-oscillating resonance frequency, quality factor, and aging of the damping layer of sensors with different thicknesses of the damping layer. The data were obtained from: five sensors with no coating (measurement of one sensor taken from [10]); six sensors coated with EIG with a composition ratio of 4:20:2 of epoxy glue, IPA, and glycerol, respectively (four of which had the same coating of 12 µm in thickness, one with a thickness of 40 µm, and one with a thickness of 127 µm); one sensor coated with EIG with a composition ratio of 4:20:1 at a thickness of 69 µm. The standard deviations of the thickness and resonance frequency given in this table refer to local variances of the thickness along the length of the beam determined by optical inspection after each deposition step and repeated measurements of $f_{0}$ over periods of several days to weeks, respectively. The average resonance frequency of all sensors with a coating thickness of $t_{2}=12\;\mu m\pm 12\;\mu m$ is $f_{0}=3061\;\mathrm{Hz}\pm 33\;\mathrm{Hz}$.

Coating Thickness, $t_{2}$	Resonance Frequency, $f_{0}$	Quality Factor, *Q*	Aging of Layer
—	3068 Hz ± 50 Hz	610 ± 106	—
12 µm ± 12 µm	3079 Hz	245	91 days
	3018 Hz	198	280 days
	3052 Hz	248	324 days
	3093 Hz	370	511 days
40 µm ± 17 µm	2674 Hz ± 10 Hz	81 ± 8	25 days to 51 days
69 µm ± 25 µm	2443 Hz	22	328 days
127 µm ± 64 µm	1523 Hz	1.5	40 days

**Table 2 sensors-23-02003-t002:** Dimensions of the silicon cantilever microprobe. The polymeric mixture deposited on top has a total volume given by the product of the average thickness of the coating, the width of the beam, and the length of the beam. Despite optically measured non-uniform coating thicknesses, the standard deviation of the resonance frequency measured with the four devices of the same average thickness was as low as 1.1% (cf. Table 1), indicating the reasonably controllable character of the coating.

Parameter		Symbol	Value
Length		*L*	5 mm
Width		*w*	200 μm
Thickness		$t_{1}$	50 μm
Tip height	silicon	*h*	100 μm
	diamond		210 μm
Tip mass	silicon	*M*	0.15 μg
	diamond		12 μg
Tip radius	silicon	*R*	0.1 μm
	diamond		2 μm
Tip cone angle	silicon	β	40°
	diamond		90°

**Table 3 sensors-23-02003-t003:** Initial and reduced quality factor, $Q_{\mathrm{ini}}$ and $Q_{\mathrm{red}}$, respectively, of the fundamental vibration mode using different damping approaches reported in the given literature sources. For the comparison of the effectiveness of damping, the right column lists the respective reduction ratio $Q_{\mathrm{red}}/Q_{\mathrm{ini}}$.

Approach	Initial $Q_{\mathrm{ini}}$	Reduced $Q_{\mathrm{red}}$	$Q_{\mathrm{red}}/Q_{\mathrm{ini}}\times 10^{4}$
Closed-loop control with external sensor feedback			
Photothermal actuation [13]	1800	2	11
Radiation pressure [14]	137,000	55	4.0
Magnetic force [15]	2013	5	25
External piezoactuator [16,17,18,19,20]	44,200	7.8	1.8
Integrated piezoactuator [21,22,23]	—	—	71 *
Closed-loop control with integrated sensor feedback			
Integrated piezoactuator [24,25,26,27]	226	17	752
Open-loop control			
Additional (piezo) actuator [28]	8746	5533	6326
Optical pumping [29,30,31]	259	28	1065
Mechanical pumping [32,33]	4599	230	500
Passive damping			
Piezoelectric shunt [34]	297.6	35.5	1193
Viscous coating [35,36]	502 ± 8	398 ± 8	7928 ${}^{†}$
High-loss-material cantilever [37]	350 ${}^{‡}$	21	600
This work	610 ± 106	1.5	25

∗ Ratio of the peak amplitudes [21]. ^†^ The ratio of *Q*_red_/*Q*_ini_ = 1961 × 10^−4^ was achieved with a cantilever of a different geometry [36]. ^‡^ Q factor of a silicon cantilever with a resonant frequency similar to that of the high-loss-material cantilever [37].

## Data Availability

The data presented in this study are openly available on Zenodo at https://doi.org/10.5281/zenodo.7627017.

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
