# Peer review of "Damped Cantilever Microprobes for High-Speed Contact Metrology with 3D Surface Topography"

_sensors, 2023, doi:10.3390/s23042003_

Round 1
Reviewer 1 Report
Please check the content in the attachment.

Reviewer 2 Report
Dear authors,
the presented paper is well structured and clearly presents the application of the developed sensor.
However, there is no clearly presented contribution to the science of your work - what makes your sensor better and different from similar ones? (how to present your sensor consider e.g. https://doi.org/10.5545/sv-jme.2022.275)
In your presentation, I am also missing a more extensive state of the art of such sensors (you have last but not least 22 references only) in the first part of the paper. Improve this section and add the description of the comparative sensors, then explain where and how much your is better.
Reviewer 3 Report
The present article titled, "Damped cantilever microprobes for high-speed contact metrology with 3D surface topography" presented the fabrication of cantilever microprobes using common viscoelastic materials such as epoxy glue, IPA and glycerol. The work is of high quality and results are well explanatory. Whereas no specific reason is assigned why ratio of glue to IPA to glycerol of 4:20:2 or 4:20:1 is maintained during cantilever fabrication. The authors should try to explain the idea vividly and any specific reason should be proven experimentally.
